# Cosmogenic Background Suppression at ICARUS †

**Biswaranjan Behera** ‡,§ 

Department of Physics, Colorado State University, Fort Collins, CO 80523, USA; bbehera@fnal.gov
† Presented at the 23rd International Workshop on Neutrinos from Accelerators, Salt Lake City, UT, USA, 30–31 July 2022.
‡ Current address: Department of Physics, University of Florida, Gainesville, FL 32611, USA.
§ For the ICARUS Collaboration.

**Abstract:** The ICARUS detector will search for LSND-like neutrino oscillations exposed at shallow depths to the FNAL BNB beam, acting as the far detector in the short-baseline neutrino (SBN) program. Cosmic background rejection is particularly important for the ICARUS detector due to its larger size and distance from neutrino production compared to the near detector SBND. In ICARUS, the neutrino signal over the cosmic background ratio is 40 times more unfavorable compared to SBND, partly due to an out-of-spill cosmic rate that is over three times higher. In this paper, we will illustrate techniques for reducing cosmogenic backgrounds in the ICARUS detector with initial commissioning data.

**Keywords:** neutrino; cosmogenic backgrounds; LArTPC; cosmic ray tagger



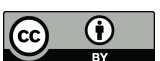

## 1. Introduction

The imaging cosmic and rare underground signals (ICARUS) [1] detector at Fermilab is based on a liquid argon time projection chamber (LArTPC) technology. ICARUS was refurbished to detect neutrinos generated in Fermilab's booster neutrino beamline (BNB) as part of the short baseline neutrino (SBN) program. In addition, ICARUS is exposed to off-axis neutrinos generated from the main injector (NuMI) beam. The basic difference is that BNB produces low-energy neutrinos compared to NuMI neutrinos and this is in the energy range of the future DUNE experiment. As an LArTPC, important components of the detector are the time projection chamber (TPC) and photon detection system. When neutrinos from the booster beam interact in the liquid argon, they give off charged particles that ionize the argon. This ionization charge is detected directly by directing free electrons to wire electrodes within milliseconds. Additionally, the excitation of argon molecular states generates ultraviolet scintillation light, which reaches sensors on the wall very quickly, within nanoseconds, leading to indirect detection of the charge, and providing an accurate measurement of the time of the event. The wire electrodes and light sensors work in concert to enable the three-dimensional reconstruction of events in the detector. The ICARUS cosmic ray tagger (CRT) system consists of three subsystems, the top, side, and bottom, which surround the TPCs to reject cosmogenic activities in the detector. The top part features newly made plastic scintillator modules covering 400 m² above the cryostat and intercepts about 80% of the cosmogenic muon flux. The top module consists of two types: the top vertical module, which covers the rim region, and the top horizontal module, which covers the top roof (Figure 1 (left)). The bottom portion utilizes Double Chooz modules, initially built for the Double Chooz experiment. These modules are placed below the cryostat. The side CRT utilizes recycled modules from the decommissioned MINOS detector, providing coverage for the sides of the cryostat and reducing 20% of the intercepted cosmic flux. However, the coverage on the north side is slightly reduced due to space constraints in the building. The combination of these subsystems achieves 97% geometric efficiency in intercepting muons entering the cryostat, as determined by simulation studies [2].

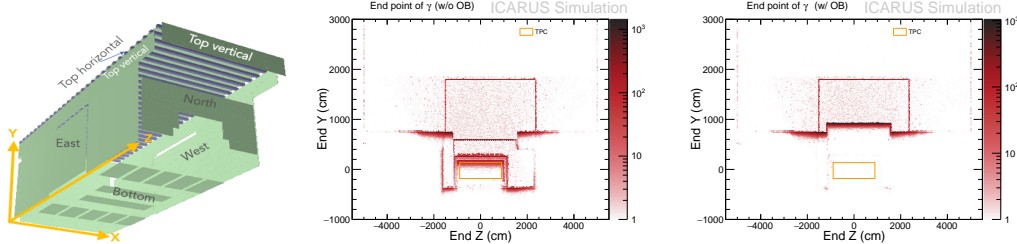

**Figure 1.** A sketch of the CRT geometry with coordinates (**left**). End point of primary neutrons and $\gamma$s as seen in a Y/Z projection, without OB (**middle**) and with overburden (**right**) (the $\nu$ beam is along the z axis). The black square shows the position of the TPCs.

To mitigate cosmic activity in the ICARUS detector, a 2.85 m thick concrete block, referred to as the overburden, has been placed above the detector. The overburden consists of three layers of concrete blocks, each approximately 1 m tall, giving a total mass of 5 million pounds. Additionally, a plastic scintillator bar detector called CRT is used to reduce cosmogenic backgrounds by surrounding the TPC with a tagger. In the approved FNAL SBN experiment, both the near and far detectors employ a $4\pi$ cosmic ray tagger (CRT) detector and a 2.85 m concrete overburden to mitigate the impact of cosmic rays. The rejection of cosmic backgrounds is especially important for the ICARUS detector, which, due to its larger size and distance from the target compared to SBND, experiences approximately five times the cosmic ray rate while the neutrino interaction rate is reduced by a factor of approximately ten.

The ICARUS T600 LAr-TPC detector will search for LSND-like neutrino oscillations exposed at shallow depths to the FNAL BNB beam, in the context of the SBN program. The SBN experiment is expected to reach $5\sigma$ sensitivity within 3 years of data-taking by comparing the neutrino spectra collected by ICARUS T600 (760 t LAr) and SBND (112 t LAr) detectors at 600 m and 110 m from the target. During its first year of operation, ICARUS will also investigate the NEUTRINO4 claim with both BNB and NuMI off-axis beams.

Any possible background source by cosmic rays mimicking $\nu_e$ CC interactions that could potentially spoil the experimental sensitivity has to be suppressed well below the unavoidable 1500 $\nu_e$ CC events expected from the intrinsic BNB electron beam component. Similar requirements apply to the NEUTRINO4 search with both BNB and NuMI beams, strongly reducing the cosmic-induced background events.

## 2. Cosmogenic Background Suppression

Since ICARUS is situated just below ground level with no earth overburden, they are exposed to a large flux of cosmic rays. Estimating the portion of this cosmic flux that enters the detector and exploring ways to reduce it are crucial to successfully performing most of the neutrino beam-related analyses in the SBN program.

We can divide cosmic particles into two categories: in-time and out-of-time cosmic particles. In-time refers to cosmic particles that enter the detector during the beam spill. On the other hand, out-of-time cosmic particles are those that cross the detector during the drift time. For the BNB beam, when factoring in the overburden in ICARUS, there is approximately 1 neutrino interaction for every 180 spills. Additionally, there is approximately 1 in-time cosmic ray for every 44 spills and 14 out-of-time cosmic rays during each TPC drift time. In the case of the NuMI beam, there is a higher presence of cosmic rays within the beam gate compared to the BNB. Specifically, for NuMI, there is approximately 1 neutrino interaction for every 15 spills, 1 in-time cosmic ray or every 10 spills, and an average of 56 out-of-time cosmic rays during each TPC drift time.

### 2.1. Using Concrete Overburden

According to the SBN proposal, which simulated the overburden using a simplified detector description in the open air, the overburden is needed, as "[...] a 3 m rock coverage reduces by a factor 400 the number of primary photons above 200 MeV in the active

volume" [1]. The 200 MeV reference is used here only because electromagnetic activity above 200 MeV is considered relevant for the main SBN analysis. Also, an overburden thickness of approximately 3 m is necessary to prevent secondary particles produced within the overburden from escaping, which would introduce further background interference.

The effect of the overburden on the cosmic particles in ICARUS is shown in this paper, using the latest available simulations, which improve upon the ones used for the SBN proposal as they include full detector modelings and building geometries.

The COsmic Ray SImulations for KAscade (CORSIKA) software [3] is used to generate cosmogenic particles in ICARUS simulations. CORSIKA simulates air showers generated from high-energy cosmic particles using the proton-only model, where only primary cosmic protons are assumed to contribute to the Earth's cosmic-ray flux. The distributions of the end point of the primary neutrons and photons, shown in Figure 1 (middle and right), also demonstrate the effect of the overburden in removing these particles.

For the $\nu_e$ analysis, an important potential source of the background is represented by the electromagnetic showers with E > 200 MeV produced inside the TPC by cosmic rays [1]. The ICARUS detector is expected to select 1500 $\nu_e$ CC interactions by the intrinsic BNB $\nu_e$ component [1] at 600 m from the target; even small additional electromagnetic background events from cosmic rays could spoil the reach of the SBN program. Similar considerations hold for the NuMI beam events.

Charged pions from cosmic neutron interactions in LAr, producing one charged pion and at least one proton, are possible sources of background events since pions in the TPC can be misidentified as muons and mimic-contained QEs $\nu_\mu$ CC. The total number of these events over three years of data collection will be 1165 (20), without overburden (with overburden), respectively [4]. This is a minor contribution with overburden. To summarize, the findings of this study [5] confirm the crucial role of the overburden in effectively reducing the cosmic background to $\nu_e$. The overburden not only decreases the direct cosmic flux by suppressing primary hadronic and electromagnetic components but also enhances the efficiency of the CRT system in rejecting any remaining cosmic contributions to the $\nu_e$ background.

The installation of the last concrete block was completed on 7 June 2022, marking the beginning of ICARUS data-taking for physics with both BNB and NuMI beams. Top CRT cosmic event rates before and after the installation of concrete overburden are shown in Figure 2 for horizontal (left) and vertical (right) modules. The mean rates for horizontal and vertical modules were approximately 610 Hz and 260 Hz, respectively, before the installation of the overburden. After the installation, the rates reduced to 330 Hz for horizontal modules and 180 Hz for vertical modules. Except for variation due to the concrete block placement above the detector, the rates are stable on a time scale of months.

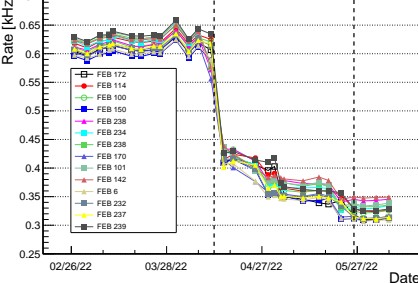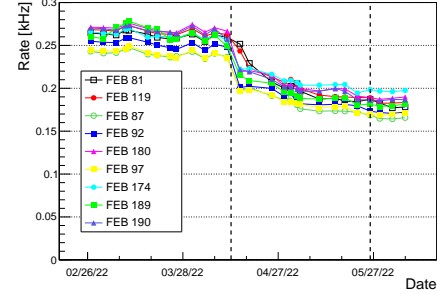

**Figure 2.** Cosmic ray rates as functions of time for a set of the top CRT horizontal (**left**) and vertical (**right**) modules. Numbers in the legend indicate the module's front-end boards and the black dot lines indicate the beginning and the end of 3 m overburden installation over the displayed modules. The rates were reduced from approximately 610 Hz to 330 Hz for horizontal modules and from 260 Hz to 180 Hz for vertical modules after the installation of the overburden.

Hardware-wise, the reduction of fluxes by the overburden and the suppression of soft cosmic ray components reduce the probability of multiple particle hits over the same CRT counter, resulting in better CRT tagging and timing performances.

The $\gamma$-initiated showers from $\pi^0$ generated by cosmic hadrons cannot be rejected using the CRT. They represent the dominant background sources, which can be strongly suppressed by the overburden. The remaining $\gamma$-initiated showers, produced from muon through bremsstrahlung and $\pi^0$ nuclear photo-production, are effectively rejected by recognizing the muon with the CRT or the TPC.

*2.2. Using Cosmic Ray Tagger (CRT)*

The CRT system detects charge particles entering the detector from the outside, whose tracks may interfere with the reconstruction of beam neutrino events. The CRT system surrounds the exterior of the warm vessel as much as possible. It has three subsystems with different modules and readout electronics. The early commissioning results from the CRT are illustrated in [2]. The CRT system is expected to strongly mitigate the events associated with primary muons entering the detector, while the events induced by cosmic primary neutrons can only be suppressed by the overburden. In particular, the showers initiated by $e^{\pm}$ generated by muons via ionization or pair production are rejected by observing the muon signal in the CRT or inside the TPC.

2.2.1. Using Association between the TPC Track and CRT Hit

If a particle crosses the detector before or after the trigger time, the TPC-reconstructed x-position will be shifted. The time of the TPC track can be found by matching with CRT hits using the algorithm described below:

- Take each TPC track and find the allowed time frame that will keep the track inside the TPC.
- For each CRT hit in this time range, calculate the distance of the closest approach (DCA) using the start and end directions.
    - For each CRT hit, displace the track by -vt along the x-axis, where "v" represents velocity and "t" represents time value.
    - Extrapolate the displaced track to the plane of the CRT hit.
    - Calculate the distance in the plane from the track intercept to the CRT hit. Find the track–CRT pairing that has the smallest distance.
    - If this distance is <30 cm, then they are matched. Assign the time of the CRT hit as the t0 of the TPC track.
- There are few filtrations applied to the DCA calculation.
    - Accept the TPC track length > 20.0 cm and the PE value of the CRT hit > 60.
    - Keep the maximum uncertainty on the CRT hit to 20 cm.

The algorithm's validation was conducted, and the distribution of the distance of the closest approach (DCA) for all tracks, including cathode crossing tracks, is shown in Figure 3. Notably, there are peaks at 12 to 15 cm, indicating that scattering tracks exhibit larger DCA values. In this scenario, the efficiency is low, but the purity is high. To identify the best match, a cut is applied to the closest distance between the CRT hit and TPC track, requiring it to be less than 30 cm. These tracks can be identified as cosmics entering the detector, and by applying this cut, we can effectively filter out cosmic rays. Currently, we are progressing with this framework and continuously improving the algorithms to enhance cosmic rejection.

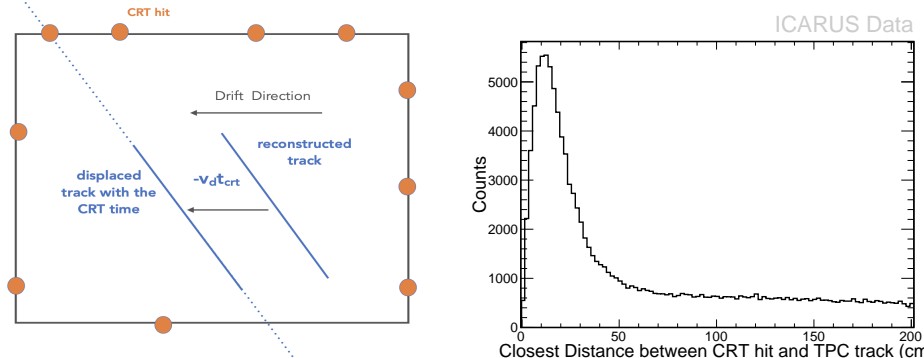

**Figure 3.** The figure portrays the method of the CRT hit and TPC track association (**left**). The distance of the closest approaches (cm) for all tracks (**right**).

### 2.2.2. Using Time of Flight (TOF) between Light and CRT System

Considering that both PMT and CRT systems will achieve ∼1 ns of level time resolution in ICARUS, it is feasible to use the time-of-flight veto method, which uses the reconstructed information from the light system and CRT system to reject the cosmic particle. The time of flight is calculated by the delay between the CRT hit and the first PMT signal. If a cosmic particle enters from the top of the detector, the time will be first registered by the CRT system and then the PMT sub-system in ICARUS. Therefore, an event with a negative time difference between CRT and PMT will be considered as cosmic. However, the muon existing from the neutrino interaction can be referred to as time-of-flight positive. The CRT timing system was synchronized with the light system, using the common trigger signal recorded by the CRT and light system. A preliminary calculation of the TOF for cosmic muons was performed by selecting particles entering the top CRT modules and generating a flash in the active argon volume. The preliminary distribution of the time differences between top CRT hits and PMT signals is shown in Figure 4. The measured average TOF of 24 ± 9 ns is in agreement with the expected ∼26 ns evaluated from the distance between the top CRT plane and the first PMT row.

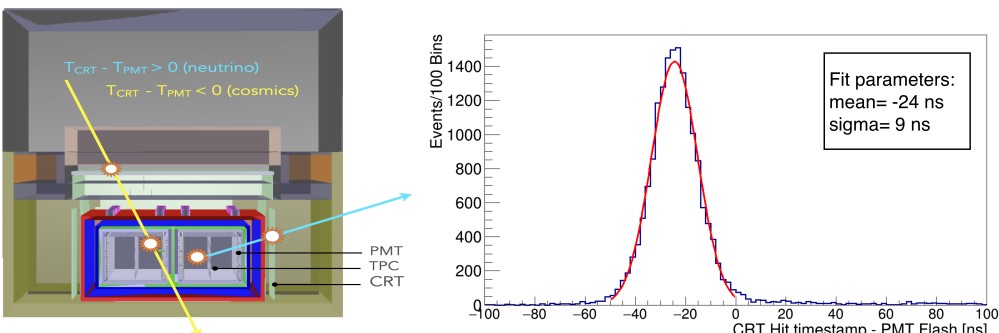

**Figure 4.** The figure depicts the method of the CRT hit and PMT flash matching (**left**). The quantity "$T_{CRT} - T_{PMT}$" represents the time difference between the arrival of a signal at the cosmic ray tagger (CRT) and the corresponding signal at the photomultiplier tube (PMT). This time difference provides valuable information for distinguishing between different types of tracks in the detector. The time difference between CRT hits (from top CRT) and PMT flashes using the BNB spill (**right**).

### 2.3. Using TPC Alone

The effectiveness of TPC in rejecting cosmic tracks has been demonstrated. The analysis of the tracks shown in Figure 5 reveals that Tracks 1, 2, 3, and 4 are out-of-time tracks, reconstructed outside the physical drift window. Track 5 exhibits a top-to-bottom traversal across the detector, while Track 6 enters the detector from the top and exits through the wire planes. These observations provide valuable insights into the presence

and characteristics of cosmic tracks in the data analysis. The robust reconstruction capability of TPC allows us to effectively reject various types of cosmic tracks.

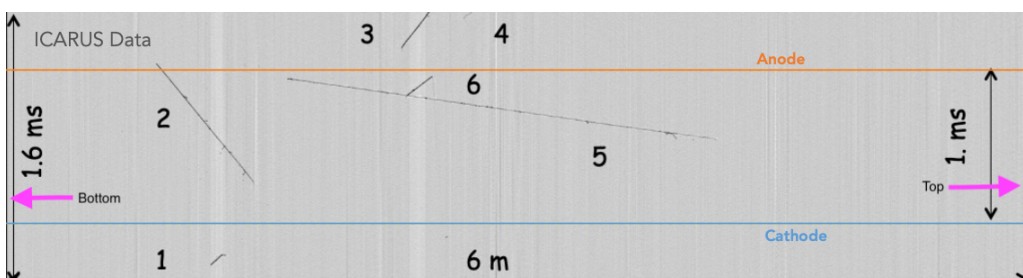

**Figure 5.** This is a real event taken in September 2021. Here, each track is of cosmic origin. The orange lines represent the anode, while the blue lines signify the cathode. The space between these two lines is referred to as the drift window.

## 3. Summary

In this paper, we highlighted key techniques used for rejecting cosmogenic backgrounds. It is important to note that there are additional methods that have not been discussed, including the utilization of stopping muons, proton bunch structures, and more. Currently, we are actively developing and validating various algorithms to reject cosmics by using commissioning data.

**Funding:** This research was funded by the US Department of Energy, award DE-SC0017740.

**Institutional Review Board Statement:** Not applicable.

**Informed Consent Statement:** Not applicable.

**Data Availability Statement:** ICARUS Experiment at Fermilab Research Data Policies at https://icarus-exp.fnal.gov (accessed on 1 August 2022).

**Acknowledgments:** We kindly acknowledge the assistance and encouragement received from ICARUS and SBND collaborators in particular D. Gibin, C. Farnese, R. Wilson, and M. Stancari. We appreciate the time and effort made by the ICARUS editorial board for carefully reading and providing feedback for this article.

**Conflicts of Interest:** The author declares no conflict of interest.

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
