# Peer review of "Cosmogenic Background Suppression at ICARUS†"

_psf, doi:10.3390/psf8010055_

Round 1
Reviewer 1 Report
The structure could be arranged better and some critical contents could be added/supplemented, where some could be removed to avoid duplication. Here are a few example suggestions:
- Chap 1. Introduction contains detailed information about the TPC, but not enough about the CRT and overburden, which I believe is more important for this presentation.
- line 40: the last sentence of the Introduction discussing about cosmic ray rate, could be located before discussing about the physics effect (i.e. line 32-36).
- line 40: the 5 times larger cosmic rate x 10 times reduced neutrino rate makes up 50, not 40 as mentioned in the abstract. Also, the 3 times larger cosmic rate mentioned in the abstract doesn't match the number given here, which is 5 (if 5 corresponds to different definition instead of out-of-spill cosmic rate, that info could be given).
- line 52: "one in-time cosmic ray in every 44 (10) spills" seems to be the case for BNB (44) /NuMI (10) beam respectively? If so, better to explicitly write it. Then the following information of "14 out-of- time cosmic rays" does not have parenthesis for NuMI case. If it only exists for BNB, maybe "during each TPC drift time for BNB beam" can be presented earlier, instead of 'during the drift time' in line 50.
- line 53 it is unclear "factor of 4" is compared to what.
- line 55 ~ 58: this information could have been merged with the information given around line 37.
- line 69: what is the difference between the "full detector modelings" and "the building geometries"?
- Figure1 caption: the acronym "OB" is not defined
- line 87: this conclusion only covers v_e BG, not the CCQE v_mu mentioned just before.
- line 89: the contents after "in addition..." is not mentioned before, therefore hard to be contained in a concluding paragraph.
- line 91-41: this detailed information about OB could have better been given before (around line 37 or line 55).
- line 95: further information (i.e. drawing) could be given to understand what is 'Top' CRT, and what are 'horizontal' and 'vertical' modules.
- Figure 2 caption: "the black dot lines indicate the beginning and the end of 3 m overburden installation over the displayed modules" <- what are black dot lines?
- Figure 2 caption: the rates reduced from ∼ 610 (260) Hz to 330 (180) Hz before (after) the installation of overburden for horizontal and vertical modules, respectively. -> I guess "the rates reduced from ∼ 610 (260) Hz to 330 (180) Hz after the installation of overburden for horizontal (vertical) modules."
- line 101: could you be more specific what you mean by 'Hardware wise'? And could you give definition of 'soft' cosmic rays?
- line 112: what is 'warm vessel'?
- line 112: It has three sub-systems with different modules and different readout electronics. -> could you give more detailed information? It may be helpful to understand what is 'vertical module' and 'horizontal module' mentioned before.
Chap 2.2.1: it is unclear to understand the application and impact of this method on the Cosmogenic Background Suppression.
line 127: v and t are not defined.
line 146: what is PMT 'sub' system?
Figure 5: please indicate where is top and bottom.
Also the English style can be improved, i.e.:
line 16: component of the detector is -> components of the detector are
line 35: requirements ... to strongly reduce -> requirements ... of strongly reducing
line 50: However -> On the other hand,
line 52: however -> however, there is
line 54: cosmic -> cosmics, or cosmic rays
line 57: in addition,
line 77 and after: e.m. -> electromagnetic
line 82: "Charged pion from cosmic neutron :" -> remove
line 85: Total number of events -> Total number of these events
line 85: in 3yr of data taking -> over a period of three years of data collection
line 85: will 1165 (20) in -> will be 1165 (20)
line 91: is consists of -> consists of
line 94: Before and after the installation of concrete overburden Top CRT cosmic event rates -> Top CRT cosmic event rates before and after the installation of concrete overburden
Figure 2 caption: Cosmic Rays rates: Cosmic ray rates:
line 101: Hardware wise -> Hardware-wise,
Figure 3: the figure is too small to read the text
line 120: TPC reconstructed x- position -> TPC-reconstructed x- position
line 141: Considering both PMT and CRT system will achieve ∼ 1 ns level time resolution in ICARUS. Therefore, it is feasible that the time-of-flight veto method will be used to reject the cosmic particle from the neutrino interaction in the detector using the reconstructed information from light system and CRT system. -> Considering that both PMT and CRT system will achieve ∼ 1 ns level time resolution in ICARUS, it is feasible to use time-of-flight veto method, which uses the reconstructed information from light system and CRT system, to reject the cosmic particle.
line 145: If a cosmic particle entering from the top of detector the time will first registered by CRT system and then the PMT sub system in ICARUS. -> If a cosmic particle enters from the top of detector, the time will be first registered by CRT system and then the PMT sub system in ICARUS.
line 147: Therefore the time difference between CRT and PMT will be negative and considered as cosmics. -> Therefore, an event with negative time difference between CRT and PMT will be considered as cosmics.
Figure 4 caption: The figure depicts the method of CRT hit and PMT flash matching (left). -> can there be more description about Tcrt, Tpmt, etc.? Also texts are too small to read.
line 154: Figure 4, the measured -> Figure 4. The measured
line 158: Track that has signals on the wires in the nonphysical drift window. -> ??
line 160: From the Fig 5, we observed several cosmic tracks and listed as out of time tracks are reconstructed outside the physical drift window (Track 1, 2, 3, 4). -> From the Fig 5, we observe several cosmic tracks. The ones reconstructed outside the physical drift window are listed as out of time tracks are (Track 1, 2, 3, 4).
Figure 5 caption: The tracks entering from the top and 162 exiting from the wire planes (Track 6). -> ???
